# Adversarial Generative Flow Network for Solving Vehicle Routing Problems

**Ni Zhang**[1], **Jingfeng Yang**[2], **Zhiguang Cao**[1,*] **Xu Chi**[2]
[1]Singapore Management University, Singapore
[2]Singapore Institute of Manufacturing Technology (SIMTech),
Agency of Science, Technology and Research (A*STAR), Singapore
`zhangni0128@outlook.com, Yang_Jingfeng@simtech.a-star.edu.sg`
`zgcao@smu.edu.sg, cxu@simtech.a-star.edu.sg`

## Abstract

Recent research into solving vehicle routing problems (VRPs) has gained significant traction, particularly through the application of deep (reinforcement) learning for end-to-end solution construction. However, many current construction-based neural solvers predominantly utilize Transformer architectures, which can face scalability challenges and struggle to produce diverse solutions. To address these limitations, we introduce a novel framework beyond Transformer-based approaches, i.e., Adversarial Generative Flow Networks (AGFN). This framework integrates the generative flow network (GFlowNet)—a probabilistic model inherently adept at generating diverse solutions (routes)—with a complementary model for discriminating (or evaluating) the solutions. These models are trained alternately in an adversarial manner to improve the overall solution quality, followed by a proposed hybrid decoding method to construct the solution. We apply the AGFN framework to solve the capacitated vehicle routing problem (CVRP) and the travelling salesman problem (TSP), and our experimental results demonstrate that AGFN surpasses the popular construction-based neural solvers, showcasing strong generalization capabilities on synthetic and real-world benchmark instances. Our code is available at `https://github.com/ZHANG-NI/AGFN`.

## 1 Introduction

The vehicle routing problem (VRP) represents a fundamental and intricate combinatorial optimization challenge with extensive real-world implications (Toth & Vigo, 2014), including supply chain management (Lee et al., 2006), last-mile delivery services (Koç et al., 2020), and public transportation (Hassold & Ceder, 2014). Given its widespread occurrence across numerous domains, the VRPs have been the subject of extensive research for decades within the Operations Research (OR) community. Particularly, practitioners employ both exact and heuristic methods to tackle complex optimization problems including VRPs. Exact methods, such as branch-and-bound (Lawler & Wood, 1966), branch-and-cut (Tawarmalani & Sahinidis, 2005), and column generation (Barnhart et al., 1998), guarantee optimal solutions but often face computational limitations for large-scale instances. Consequently, heuristic approaches like tabu search (Osman, 1993), adaptive large neighborhood search (Ropke & Pisinger, 2006), and hybrid genetic search (Vidal, 2022) have gained prominence for their ability to efficiently produce near-optimal solutions.

While these traditional methods continue to play a vital role, recent years have seen the rise of learning-based Neural Combinatorial Optimization (NCO) approaches that can learn solution construction policies directly through supervised or reinforcement learning, without relying on much problem-specific heuristic design. However, these neural constructive solvers encounter substantial challenges concerning scalability. Typically based on Transformer architectures, their training becomes increasingly difficult as problem sizes grow. Consequently, while these approaches exhibit strong performance on small-scale problems, they often struggle to generalize effectively to larger and more complex real-world instances. To address these generalization issues, recent studies have

---

*corresponding author

proposed several innovative approaches. Luo et al. (2023) introduced the Light Encoder and Heavy Decoder (LEHD) Transformer-based model via supervised learning, which enhances the model's generalization ability. Xin et al. (2022) proposed to enhance the generalization by generating adversarial instance distributions specifically designed to be challenging for neural constructive models to solve. However, both methods still rely on the existing Transformer-based architecture, which remains difficult to be trained directly on (relatively) large instances due to limited device memory.

In this paper, we aim to develop a new constructive neural solver that does not rely on the Transformer architecture. Inspired by recent advancements in Generative Flow Networks (GFlowNets) (Bengio et al., 2021; 2023; Zhang et al., 2024) for solving COPs, we propose a novel GFlowNet-based neural VRP solver with adversarial training. Specifically, we leverage the GFlowNet to act as a generator, with the objective of constructing diverse solutions using a forward sampling policy, and a discriminative network classifier to evaluate the quality of the generated solutions. The GFlowNet and the discriminative network are trained alternately in an adversarial manner. In particular, the GFlowNet is trained by minimizing the trajectory balance objective function (Malkin et al., 2022) while the discriminative classifier is trained to distinguish between the original solutions produced by the GFlowNet and the enhanced solutions obtained through a local search. This iterative process enables the GFlowNet to progressively generate higher-quality solutions based on feedback from the discriminator. Moreover, to further leverage the inherent diversity of GFlowNet, we propose a hybrid decoding method that combines greedy and sampling schemes to construct the routes more effectively. In summary, our contributions are outlined as follows:

- We propose a constructive Adversarial Generative Flow Networks (AGFN) framework for solving vehicle routing problems including CVRP and TSP in an end-to-end manner.
- We introduce a simple yet effective hybrid decoding method that significantly improves the solution quality with a modest increases in inference time.
- We perform a comprehensive evaluation of our AGFN, and experimental results on both synthetic and benchmark instances confirm its competitiveness against traditional and neural baselines, with effective generalization to varying problem sizes and distributions.

## 2 RELATED WORKS

**Neural Solvers for VRPs.** The literature on neural solvers for vehicle routing problems (VRPs) can be broadly categorized into two main approaches: (1) the **construction-based** method, and (2) the **improvement-based** method. Kool et al. (2019) first introduced an Attention Model (AM) using a Transformer architecture to solve VRPs. The Policy Optimization with Multiple Optima (POMO) proposed by Kwon et al. (2020) further enhanced the AM model by employing a more advanced learning and inference strategy that leverages multiple optimal policies. Building upon AM and POMO, numerous other construction-based solvers have been developed (Kwon et al., 2021; Li et al., 2021a; Xin et al., 2020; 2021a; Chalumeau et al., 2023; Luo et al., 2023). Compared to traditional handcrafted heuristics, these neural solvers employing Transformer architectures can generate solutions quickly. However, these methods often face scalability challenges due to the quadratic complexity of the self-attention mechanism, which makes them resource-intensive for training and limits their generalizability to large problem instances. On the other hand, **improvement-based** neural solvers iteratively refine solutions by combining with traditional heuristic search algorithms, such as beam search (Choo et al., 2022), ant colony optimization (Ye et al., 2024), local search (Hudson et al., 2022), and dynamic programming (Kool et al., 2022). Other notable works include Li et al. (2018); Chen & Tian (2019); Lu et al. (2019); Hottung et al. (2021). Generally, improvement-based methods can produce superior results when given additional inference time compared to their construction-based counterparts, but also suffer from scalability issue.

**GFlowNets for Combinatorial Optimization.** GFlowNets (Bengio et al., 2021) are probabilistic models designed to generate diverse solutions (structures or sequences) by modeling a distribution proportional to a specified reward function. The first notable application of GFlowNets was in molecular design for drug discovery, a problem often approached as a black-box optimization challenge. Very recent studies have applied GFlowNets to various combinatorial optimization problems (COPs), such as the maximum independent set (Zhang et al., 2023), job scheduling (Zhang et al.), and vehicle routing problems (Kim et al., 2024b), due to their strong capability to effectively explore vast, discrete solution spaces while balancing exploration and exploitation. While

GFlowNet is highly capable of generating diverse solutions, it often results in over-exploration, leading to being trapped in numerous locally optimal solutions with low-reward. To overcome this issue, Kim et al. (2023) proposed local search GFlowNets (LS-GFN) to enhance the training effectiveness by encouraging the exploitation of high-rewarded solution spaces. Similarly, Kim et al. (2024b) integrated GFlowNets with ant colony optimization, using a local search operator to manage the trade-off between exploration and exploitation in solving combinatorial optimization problems.

To the best of our knowledge, the most closely related work utilizing GFlowNet for solving VRPs is Kim et al. (2024b), where GFlowNet is trained to learn a constructive policy that provides an informed prior distribution over the edges of routes, guiding the search of the ant colony optimization (ACO). In contrast, our proposed framework enables GFlowNet to directly generate high-quality solutions, without relying on additional heuristic algorithms like ACO during inference.

## 3 ADVERSARIAL GFLOWNET

One of the distinctive features of GFlowNet, compared to traditional models for learning to construct solutions, is its ability to generate a diverse range of solutions—not only optimal ones but also suboptimal and even inferior ones. Previous research has primarily focused on training models to produce highly diverse solutions by incorporating diversity reward functions and other techniques (Nica et al., 2022; Jain et al., 2022). However, there has been relatively little exploration of how the multiple solutions generated by GFlowNet during training can be utilized to enhance performance beyond merely increasing diversity. In solving COPs with GFlowNet, most existing studies have emphasized high-reward solutions while paying insufficient attention to low-reward ones (Zhang et al., 2023; Shen et al., 2023; Kim et al., 2023; 2024b;a) to avoid over-exploration. This imbalance, however, can cause certain edges in the graph of a VRP instance to be overestimated by the GFlowNet, increasing the risk of local optima traps, which ultimately affects the overall performance.

To address this issue, we propose Adversarial GFlowNet (AGFN) for solving VRPs. It leverages the inherent diversity brought by GFlowNet and incorporates adversarial training to evaluate the quality of the generated solutions, further balancing exploration and exploitation. With the adversarial scoring mechanism, we provide more nuanced feedback to GFlowNet, which refines the training process. This approach directs the model to conduct a more precise evaluation of each edge while promoting a balanced representation of the entire graph for a VRP instance, thus enhancing its overall performance and generalization capabilities. Consequently, AGFN not only preserves the desired diversity brought by GFlowNet but also systematically integrates it into the optimization process. To facilitate a better understanding of our AGFN framework, we include a section that provides the preliminaries of GFlowNet in the Appendix A.

### 3.1 GENERATOR

In the GFlowNet based generator, to reduce the computational complexity, the model employs a sparsification technique on the input instance. Taking CVRP as an example, we use graph $\mathcal{G} = (\mathcal{V}, \mathcal{E})$ to represent it, where $\mathcal{V} = \{v_0, v_1, \ldots, v_n\}$ denotes the locations of all nodes, with $v_0$ representing the depot and $\{v_i\}_{i=1}^n$ representing customers. The set $\mathcal{E}$ includes each edge $e_{ij}$, associated with a travel cost $c_{ij}$ (e.g., distance). Drawing inspiration from NeuroLKH (Xin et al., 2021b), we acknowledge that handling a fully connected graph for large-scale VRP instances is computationally prohibitive. To overcome this challenge, the underlying graph is reformulated where each node is constrained to retain only $k$ of its shortest outgoing edges. In doing so, it helps reduce the computational burden while preserving the core structural features, allowing the method to scale effectively to larger problem sizes. The resulting sparse graph $\mathcal{G}^*$, consisting of the node set $\mathcal{V}$ and the sparse edge set $\mathcal{E}^*$, is then encoded into a higher-dimensional space by linearly projecting the node coordinates $\mathbf{x}_v \in \mathbb{R}^2$ and edge distances $\mathbf{x}_e \in \mathbb{R}$ into node feature vectors $\mathbf{h}_i^0 \in \mathbb{R}^d$ and edge feature vectors $\mathbf{e}_{ij}^0 \in \mathbb{R}^d$ for $i \in \mathcal{V}$ and $(i, j) \in \mathcal{E}^*$, where $d$ is the feature dimension. The selection probabilities for each edge are then computed using a Graph Neural Network (GNN) model. The GNN updates the node features $\mathbf{h}_i^{l+1}$ and the edge features $\mathbf{e}_{ij}^{l+1}$ based on the features from the $l^{th}$ layer, $\mathbf{h}_i^l$ and $\mathbf{e}_{ij}^l$, respectively. The GNN operates as follows:

$$\mathbf{h}_i^{l+1} \leftarrow \mathbf{h}_i^l + \text{ACT}(\text{BN}(\mathbf{U}^l \mathbf{h}_i^l + \mathcal{A}_{j \in \mathcal{N}_i}(\sigma(\mathbf{e}_{ij}^l) \odot \mathbf{V}^l \mathbf{h}_j^l))), \tag{1}$$

$$\mathbf{e}_{ij}^{l+1} \leftarrow \mathbf{e}_{ij}^l + \text{ACT}(\text{BN}(\mathbf{P}^l \mathbf{e}_{ij}^l + \mathbf{Q}^l \mathbf{h}_i^l + \mathbf{R}^l \mathbf{h}_j^l)). \tag{2}$$

Here, $\mathbf{U}^l, \mathbf{V}^l, \mathbf{P}^l, \mathbf{Q}^l, \mathbf{R}^l \in \mathbb{R}^{d \times d}$ are trainable parameters, ACT denotes the activation function, BN stands for batch normalization, $\mathcal{A}_{j \in \mathcal{N}_i}$ represents the aggregation operation over the neighbors of node $i$, $\sigma$ is the sigmoid function, and $\odot$ indicates the Hadamard product. The activation function (ACT) used for all layers is SiLU (Elfwing et al., 2018), while the aggregation function $\mathcal{A}$ is defined as mean pooling. To produce the edge probability distribution (heatmap) $\eta(\mathcal{G}^*, \boldsymbol{\theta}_{generator})$ using GFlowNet, the node and edge embeddings from the final layer are passed through a fully connected multi-layer perceptron (MLP), where SiLU is applied as the activation function for all layers except the last one, which employs a sigmoid function to yield normalized outputs.

Note that, the GNN used here efficiently handles the complex relationship between node and edge, and its low computational complexity makes our GFlowNet based model more effective than the Transformer one. Furthermore, this GNN has fewer parameters compared to the classic GCN (Joshi et al., 2020), allowing it to be directly trained on large problems. Later, multiple paths, represented as $\mathcal{T} = \{\tau_0, \tau_1, \ldots, \tau_K\}$, are generated through sampling and evaluated by a discriminator to obtain scores. These scores are then incorporated into the reward function as follows,

$$-\log \widetilde{R}(\tau_k) = (1 - \mathcal{S}(\tau_k)) + R(\tau_k) - \frac{1}{K} \sum_{t=1}^{K} R(\tau_t), \tag{3}$$

where $R(\tau_t)$ represents the total length of the path $\tau_t$, and a lower $R$ value indicates a higher path quality (since we aim to minimize the route length in VRPs); The scores, $\mathcal{S} \in [0, 1]$, reflect the discriminator's assessment of path quality, with values closer to $1$ indicating higher quality paths and values approaching $0$ indicating lower quality paths. As training progresses, the quality of the generated paths steadily improves, the $R$ distribution shifts towards smaller values, and the deviation of each solution's length from the mean becomes increasingly smaller. Furthermore, the solutions align more closely with the discriminator's decision criteria. Consequently, $-\log \widetilde{R}(\tau_k)$ will tend to approach smaller values.

Integrating the score as a factor into the reward function guides the generator to better utilize solutions of varying quality to train the model. Additionally, using the score as a regulator also enables the model to account for suboptimal solutions while avoiding excessive bias toward the current optimal solution, thereby enhancing the exploration capability for global optima. Finally, the gradient loss, which aims at optimizing the training process of the GFlowNet is described as follows,

$$\mathcal{L}_{TB}(\mathcal{T}; \boldsymbol{\theta}_{generator}) = \frac{1}{K} \sum_{k=1}^{K} \left( \log \frac{Z_{\boldsymbol{\theta}} * P_F(\tau_k; \boldsymbol{\theta}_{generator})}{\widetilde{R}(\tau_k) * P_B(\tau_k; \boldsymbol{\theta}_{generator})} \right)^2. \tag{4}$$

Here, $P_F(\tau_k; \boldsymbol{\theta}_{generator})$ in Eq. (4) represents the forward probability of the trajectory $\tau_k = (s_0 \rightarrow s_1 \rightarrow \cdots \rightarrow s_n)$. This probability is calculated using the edge probabilities $P_F$ which is selected from heatmap $\eta(\mathcal{G}^*, \boldsymbol{\theta}_{generator})$, defined as follows,

$$P_F(\tau_k; \boldsymbol{\theta}_{generator}) = \prod_{t=1}^{n} P_F(s_t | s_{t-1}; \boldsymbol{\theta}_{generator}). \tag{5}$$

$P_B(\tau_k; \boldsymbol{\theta}_{generator})$ in Eq. (4) represents the backward probability of the trajectory $\tau_k$, calculated from the instance, defined as follows,

$$P_B(\tau_k; \boldsymbol{\theta}_{generator}) = \prod_{t=1}^{n} P_B(s_{t-1} | s_t; \boldsymbol{\theta}_{generator}). \tag{6}$$

Generally, the GFlowNet focuses on accurately evaluating the edge probabilities during training for solving VRPs. This enables the generator to more precisely assess edge probabilities and capture more nuanced graph features, leading to a deeper understanding of the underlying structure. As a result, the generator can better handle complex relationships within the graph, which significantly enhances its generalization across a wider range of tasks and diverse graph structures.

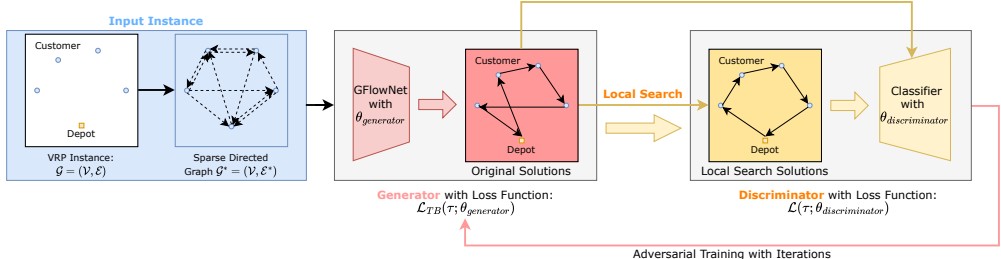

Figure 1: Illustration for the Overall Framework of Adversarial Generative Flow Network (AGFN).

## 3.2 DISCRIMINATOR

In the discriminator, the model receives two types of paths as input: the original paths $\mathcal{T}_{false} = \{\tau_0, \tau_1, \ldots, \tau_N\}$, created directly by the generator and labeled as "false", and the paths $\mathcal{T}_{true} = \{\tau_0, \tau_1, \ldots, \tau_M\}$, which are enhanced or optimized by local search and labeled as "true". The discriminator then generates scores $\mathcal{S}$ for these paths and compares the differences between the scores and their corresponding labels. For $\mathcal{T}_{true}$, the training objective is to bring the scores closer to 1, while for $\mathcal{T}_{false}$, the goal is to push the scores closer to 0. The loss function is described as:

$$\mathcal{L}(\mathcal{T}; \boldsymbol{\theta}_{discriminator}) = \frac{1}{M+N}\left(\sum_{\tau_m \in \mathcal{T}_{true}} (1 - \mathcal{S}(\tau_m))^2 + \sum_{\tau_n \in \mathcal{T}_{false}} \mathcal{S}(\tau_n)^2\right). \tag{7}$$

By doing so, the discriminator learns to differentiate between the raw generated paths and the optimized paths of higher quality, thereby enhancing its capability to evaluate the generator's output. This design not only allows the discriminator to more accurately assess the quality of the generated paths but also provides more informative feedback to the generator, leading to further improvement in the model's overall performance.

## 3.3 OVERALL FRAMEWORK

The overall framework, depicted in Figure 1, operates as a closed-loop system for adversarial learning. The generator iteratively produces new solutions based on a heatmap generated from the current model parameters and input instances, which are then evaluated by the discriminator. The discriminator assigns quality scores that are integrated into the generator's loss function, serving as feedback for backpropagation. Meanwhile, the discriminator continuously learns to distinguish subtle differences between the generated paths and the optimized paths.

The discriminator's score not only evaluates the quality of each path solution in path set $\mathcal{T}$ but also assists the generator in exploring the global optimum within the solution space. At the beginning of training, the generated paths exhibit a broad spectrum of quality, ranging from high-quality to suboptimal and even lower-quality solutions. By considering this diverse set of paths, the model converges more rapidly. As training progresses, the generator refines its ability to produce higher-quality paths, guided by the discriminator's feedback. This mechanism helps the generator focus on generating paths that increasingly align with the target distribution. The generator updates its internal parameters by minimizing the loss, further enhances its capability to produce optimal paths.

*Remarks*: In the discriminator, we employ a local search to refine the path by iteratively performing destruction, reconstruction, and top-K selection over a fixed number of rounds. It is worth noting that the local search used in the discriminator is generic, and, as shown in our subsequent experiments, alternative local search methods perform equally well. Conversely, during inference, the local search is unnecessary, as only the generator is used to construct the route.

## 3.4 HYBRID DECODING

To solve vehicle routing problems like the CVRP, traditional neural models (Li et al., 2021b; Kwon et al., 2020; Hu et al., 2020; Nazari et al., 2018) often converge to locally optimal solutions by

assigning disproportionately high selection probabilities to the optimal edges while evaluating suboptimal edges with low probabilities. Such imbalance causes these models to predominantly rely on greedy strategies for path generation, as the high evaluation bias towards optimal edges restricts the exploration of alternative paths. Hence, incorporating sampling methods may not significantly improve the performance, as they lack the flexibility to effectively explore a broader solution space.

However, GFlowNet (Bengio et al., 2023) offers a more balanced probabilistic evaluation, enabling a nuanced representation of the solution space by appropriately considering both optimal and suboptimal edges during path generation. To fully exploit this capability, we further propose a hybrid decoding method, which combines the original sampling with greedy strategies. Specifically, GFlowNet, acting as the generator, constructs each path in path set $\mathcal{T}$ by selecting the next node, $s_{t+1}$, at each step with a probability $\mathcal{P}$. This probability is used to sample the edge distribution probability $P_F(s_{t+1}|s_t; \boldsymbol{\theta}_{generator})$, derived from the heatmap and representing the likelihood of transitioning from node $s_t$ to $s_{t+1}$. With $\mathcal{P}$ serving as a hyperparameter, our model also selects the next node based on the distribution using a greedy strategy with a probability of $1 - \mathcal{P}$ during inference. The selection process is stated as follows,

$$s_{t+1} = \begin{cases} s, & \text{with probability } \mathcal{P} \\ s^*, & \text{with probability } 1 - \mathcal{P} \end{cases} \tag{8}$$

where $s_t$ denotes the current node in the trajectory, $s \sim P_F(s_{t+1}|s_t; \boldsymbol{\theta}_{generator})$, and $s^* = \arg\max_s P_F(s_{t+1}|s_t; \boldsymbol{\theta}_{generator})$. This means that with probability $\mathcal{P}$ the next node $s_{t+1}$ is chosen by sampling from the edge distribution probability $P_F(s_{t+1}|s_t; \boldsymbol{\theta}_{generator})$, while with probability $1 - \mathcal{P}$, the next node $s_{t+1}$ is selected based on the highest edge probability. On the one hand, the sampling mechanism enables the model to explore a wider range of possible paths, increasing solution diversity. On the other hand, the greedy selection ensures path quality and guides the model toward better solutions. Building on the balanced and naunced probabilistic evaluation inherent to GFlowNet, this integration allows our model to effectively utilize the strengths of both schemes, which not only expands the search space but also enhances its ability to discover higher-quality solutions by avoiding premature convergence to suboptimal regions.

## 4 EXPERIMENTS

In this section, we first conduct extensive experiments on the CVRP of various sizes to demonstrate the effectiveness of our AGFN against the traditional and neural baseline methods. Additionally, to highlight the generality of our AGFN, we also apply it to solving the TSP.

**Dataset:** Following previous works (Kim et al., 2024b; Kwon et al., 2020), we perform CVRP experiments using synthetic datasets for both training and testing. Each CVRP instance consists of a set of customer nodes, a single depot node, and a vehicle with a fixed capacity $C$. The customer nodes are characterized by their positions (2D coordinates) and demands, while the depot is defined solely by its location. To generate a random CVRP instance, the coordinates of both customers and the depot are sampled from a unit square $[0, 1]^2$, and the customer demands are drawn from a predefined uniform distribution $U[a, b]$. In our experiments, we set $a = 1$ and $b = 9$, with a fixed vehicle capacity of $C = 50$ for all problem sizes: $200$, $500$, and $1,000$ customers. Each synthetic test dataset, corresponding to $200$, $500$, and $1,000$ nodes, contains $128$ instances.

**Hyperparameters:** The number of directed edges, $k$, originating from a single node in the sparse edge set, $|\mathcal{E}^*|$, is set to $|\mathcal{V}|/4$. For training the model, we only employ the sampling decoding (without greedy selection) for route generation, with the number of sampled routes per instance, $\mathcal{N}$, set to 20. The ratio of training rounds between the generator and the discriminator is maintained at $4 : 1$. During testing, hybrid decoding is used for route generation, with $\mathcal{N}$ set to 100 and the $\mathcal{P}$ in Eq. (8), set to $0.05$ (see Appendix B for more details on the selection of $\mathcal{P}$). All experiments were conducted on a server equipped with an NVIDIA Tesla V100-32G GPU and an Intel Xeon Gold 6148 CPU. Our code is available at `https://github.com/ZHANG-NI/AGFN`.

Table 1: Overall performance comparison on the synthetic CVRP dataset. The 'Obj.' indicates the average total travel distance, while 'Time' denotes the average time to solve a single instance.

| Method | $|V| = 200$ | | | $|V| = 500$ | | | $|V| = 1000$ | | |
|---|---|---|---|---|---|---|---|---|---|
| | Obj. | Gap (%) | Time (s) | Obj. | Gap (%) | Time (s) | Obj. | Gap (%) | Time (s) |
| LKH-3(100) | 28.833135 | - | 1.65 | 66.902511 | - | 5.86 | 131.795858 | - | 19.30 |
| LKH-3(1000) | 28.278438 | -1.92 | 10.72 | 64.387969 | -3.76 | 23.55 | 124.575469 | -5.48 | 66.57 |
| LKH-3(10000) | 28.041563 | -2.75 | 59.81 | 63.320078 | -5.35 | 233.72 | 120.531406 | -8.55 | 433.90 |
| POMO(*8) | **29.178707** | **1.20** | 0.33 | 79.785673 | 19.26 | 0.88 | 192.78563 | 46.28 | 3.20 |
| POMO | 29.424647 | 2.05 | 0.26 | 83.079016 | 24.19 | 0.62 | 233.093524 | 76.86 | 1.62 |
| NeuOpt | 38.478607 | 33.45 | 17.34 | 187.812195 | 180.73 | 39.41 | - | - | - |
| GANCO | 29.978834 | 3.97 | 0.50 | 71.258026 | 6.51 | 1.31 | 145.40277 | 10.32 | 4.88 |
| AGFN-100 | 31.260145 | 8.41 | **0.17** | **71.051109** | **6.20** | **0.45** | **133.96624** | **1.65** | **0.72** |
| ACO | 71.5753186 | 143.24 | 3.50 | 187.616745 | 179.88 | 11.36 | 383.960999 | 191.11 | 25.08 |
| GFACS | 45.357657 | 57.31 | 4.82 | 76.771554 | 14.75 | 13.27 | 158.971658 | 20.26 | 28.52 |
| AGFN-200 | **30.35164** | **5.27** | **0.17** | 69.599289 | 4.03 | **0.46** | 132.477417 | 0.52 | **0.72** |
| AGFN-500 | 31.826736 | 10.38 | **0.17** | **69.375366** | **3.70** | **0.46** | **129.017487** | **-2.11** | **0.72** |
| AGFN-1000 | 32.235001 | 11.80 | **0.17** | 69.873100 | 4.44 | **0.46** | 129.624237 | -1.65 | 0.73 |

## 4.1 COMPARATIVE STUDY ON CVRP

### 4.1.1 COMPARISON OF MODEL TRAINING AT THE SAME SCALE

**Baselines:** We use LKH-3 (Helsgaun, 2000), a heuristic solver, POMO (Kwon et al., 2020), a classical construction model, NeuOpt (Ma et al., 2024), a recent improvement model, and GANCO (Xin et al., 2022), which combines adversarial training and POMO, as baselines for comparing the model trained at the same scale. All neural models, including POMO, NeuOpt, GANCO, and ours, were trained on synthetic instances with 100 nodes (since those baselines are hard to train on more than 100 nodes), and tested on 200, 500, and 1,000 nodes. For heuristic baseline LKH-3, we provide the results on 100, 1,000, 10,000 iterations, where the ones with 100 iterations are used to calculate the gaps for all other methods. Additionally, AGFN was also trained on 200, 500, and 1,000 nodes for further evaluation.

**Result:** As shown in the **upper half** of Table 1, our algorithm significantly outperforms POMO, NeuOpt, and GANCO in terms of computation time. For instances with 200 nodes, our method reduces computation time by 34.62%, 99.01%, and 66.00%, respectively, compared to POMO, NeuOpt, and GANCO. For 500 nodes, the reductions are 27.42%, 98.86%, and 65.65%. On 1,000-node instances, our method achieves a reduction of 55.55% and 85.25% in computation time compared to POMO and GANCO. Importantly, while NeuOpt demonstrates much longer computational times, both POMO and GANCO—although computationally efficient—cannot achieve the balance between runtime and solution quality that AGFN offers. For example, AGFN's computation time on instances of 1,000 nodes is only 0.72 seconds, compared to GANCO's 4.88 seconds and POMO's 1.62 seconds. This highlights AGFN's advantage in scaling to larger instances while maintaining efficiency. In terms of objective value, our model performs 18.76% better than NeuOpt on 200-node instances, but is 7.13% and 4.27% worse than POMO(*8) and GANCO, respectively. On 500-node instances, our model outperforms POMO(*8), NeuOpt, and GANCO by 8.90%, 62.17%, and 0.29%. For 1,000 nodes, our model shows a 30.51% and 7.86% improvement over POMO(*8) and GANCO, respectively. Although the generalization of our model trained on 100-node instances is slightly limited for 200-node cases, it excels at both 500 and 1,000 nodes, surpassing POMO, NeuOpt, and GANCO in terms of both computation time and objective value.

### 4.1.2 COMPARISON ON TRAINING SIZES

**Baselines:** For comparisons across different training sizes, we include the heuristic method ACO and the GFACS model (Kim et al., 2024b), which combines GFlowNet with ACO. To ensure a fair comparison of route generation capabilities between AGFN and other neural baselines, none of them in our experiments utilize local search to further refine the solution after the route is generated.

**Result:** We evaluated AGFN on instances with 200, 500, and 1,000 nodes, which are trained on datasets of corresponding sizes. The results are presented in the **lower half** of Table 1. In terms of computation time, our model surpasses ACO by 95.14%, 95.95%, and 97.09%, and GFACS by

96.47%, 96.53%, and 97.44% on 200, 500, and 1,000 nodes, respectively. This substantial reduction in runtime demonstrates the scalability of AGFN and its ability to efficiently handle larger problem instances without requiring additional heuristic search refinements during inference. Regarding objective values, our model shows improvements of 57.59% and 33.08% over ACO and GFACS on 200 nodes, 62.95% and 9.63% on 500 nodes, and 66.21% and 18.46% on 1,000 nodes. Overall, AGFN significantly outperforms both ACO and GFACS in terms of computation time and objective value on all three tested scales of the training ones. These results highlight the desirable generalization capability of our model across various problem sizes, along with its computational efficiency, making it potentially suitable for practical applications involving large-scale VRPs.

## 4.2 EXPERIMENTS ON OTHER ROUTING PROBLEM

Table 2: Overall performance comparison on the synthetic TSP dataset. The 'Obj.' indicates the average total travel distance, while 'Time' denotes the average time required to solve a single instance.

| Method | $|V| = 200$ | | | $|V| = 500$ | | | $|V| = 1000$ | | |
|---|---|---|---|---|---|---|---|---|---|
| | Obj. | Gap (%) | Time (s) | Obj. | Gap (%) | Time (s) | Obj. | Gap (%) | Time (s) |
| LKH-3(100) | 10.719512 | - | 0.57 | 16.547770 | - | 1.90 | 23.123401 | - | 4.42 |
| LKH-3(1000) | 10.657703 | -0.58 | 5.90 | 16.384766 | -0.99 | 7.19 | 22.845391 | -1.20 | 32.81 |
| LKH-3(10000) | 10.626953 | -0.86 | 40.19 | 16.304531 | -1.47 | 78.43 | 22.678438 | -1.92 | 255.05 |
| POMO(*8) | **10.894811** | **1.64** | 0.225 | 20.310184 | 22.74 | 0.59 | 32.783411 | 41.78 | 3.47 |
| POMO | 10.969289 | 2.33 | 0.13 | 20.753397 | 25.42 | 0.42 | 33.237161 | 43.74 | 1.04 |
| NeuOpt | 13.190890 | 23.05 | 6.43 | 137.858551 | 733.09 | 14.63 | 325.764008 | 1308.81 | 27.96 |
| GANCO | 11.281924 | 5.25 | 0.12 | 19.361128 | 17.00 | 0.38 | 29.989326 | 29.69 | 0.90 |
| AGFN-100 | 11.84754 | 10.52 | **0.10** | **19.076220** | **15.28** | **0.30** | **27.144802** | **17.39** | **0.75** |
| ACO | 48.3356921 | 350.91 | 1.82 | 151.4268301 | 815.09 | 5.99 | 317.4939855 | 1273.04 | 13.48 |
| GFACS | 13.448596 | 25.46 | 3.21 | 23.301532 | 40.81 | 9.86 | 36.784744 | 59.08 | 21.68 |
| AGFN-200 | **11.923518** | **11.23** | **0.10** | **18.663813** | **12.78** | **0.30** | 26.572329 | 14.92 | 0.77 |
| AGFN-500 | 12.121796 | 13.08 | **0.10** | 18.855522 | 13.95 | **0.30** | **26.529234** | **14.73** | **0.75** |
| AGFN-1000 | 12.353358 | 15.24 | **0.10** | 19.301237 | 16.64 | 0.31 | 27.144802 | 17.39 | **0.75** |

We also evaluate the performance of AGFN on the Traveling Salesman Problem (TSP). Similar to the CVRP experiments, we use LKH-3 (Helsgaun, 2000), POMO (Kwon et al., 2020), NeuOpt (Ma et al., 2024), and GANCO (Xin et al., 2022) as baselines for model training with 100 nodes, and ACO and GFACS (Kim et al., 2024b) as baselines for comparisons across different training sizes. The test datasets consist of synthetic instances at scales of 200, 500, and 1,000 nodes, with each dataset containing 128 TSP instances. Each instance comprises a set of nodes represented by 2D coordinates, which are randomly sampled from a unit square $[0, 1]^2$.

As shown in the **upper half** of Table 2, AGFN achieves significant reductions in computation time and improvements in objective value compared to POMO, NeuOpt, and GANCO on 500-node instances. Specifically, it reduces computation time by 49.15%, 97.95%, and 11.24%, while improving the objective value by 8.08%, 86.16%, and 1.47%, respectively. For 1,000-node instances, our model further reduces computation time by 27.88%, 97.32%, and 16.67%, and enhances the objective value by 18.33%, 99.87%, and 9.49% compared to the baselines. This indicates that our model performs better than POMO, NeuOpt, and GANCO in both computation time and objective value at larger scales (500 and 1,000 nodes). However, for 200-node instances, while our model achieves time reductions of 23.08%, 98.44%, and 16.67%, it slightly underperforms POMO and GANCO in terms of objective value. When comparing the performance of the three training sizes, as shown in the **lower half** of Table 2, AGFN consistently outperforms both ACO and GFACS. On 200 nodes, it reduces computation time by 94.51% and 96.88%, and improves the objective value by 75.33% and 11.34%, respectively. On 500 nodes, the reductions in computation time are 94.99% and 96.96%, with objective value improvements of 87.55% and 19.08%. On 1,000 nodes, our model reduces computation time by 94.43% and 96.54%, and improves the objective value by 91.45% and 26.21%. Overall, AGFN demonstrates superior performance compared to ACO and GFACS in computation time and objective value on all the three tested scales of the training ones.

## 4.3 GENERALIZATION ANALYSIS

We assess the generalization performance of models trained on 200, 500, and 1,000 nodes, and tested on sizes different from the training one on synthetic datasets. Additionally, we also evaluate

the performance of AGFN trained on synthetic datasets with 100 nodes, as well as the POMO and GFACS models, on the CVRPLib (Uchoa et al., 2017) and TSPLib (Reinelt, 1991) benchmarks.

The results of cross-size evaluation on synthetic datasets are also included in the **lower half** of Table 1 and Table 2. Models trained on respective sizes demonstrate strong generalization across different scales, without significant performance degradation. For real-world benchmark datasets, detailed results are shown in Table 3. On TSPLib, our model achieves an improvement in performance by 15.10% and 36.96%, while reducing computation time by 82.61% and 77.78%, compared to POMO and GFACS, respectively. On CVRPLib, our model shows a 12.61% and 39.34% enhancement in performance and a 73.06% and 89.64% reduction in computation time. Overall, AGFN significantly outperforms POMO and GFACS across the TSPLib and CVRPLib datasets. In the Appendix C, we further show that AGFN substantially outperforms other construction-based neural methods on much larger instances with up to 10,000 nodes.

Table 3: Overall performance comparison on the CVRPLib and TSPLib. The 'Obj.' indicates the average total travel distance, while 'Time' denotes the average time to solve a single instance.

| CVRPLib | Optimal | AGFN-100 | POMO(*8) | GFACS | TSPLib | Optimal | AGFN-100 | POMO(*8) | GFACS |
|---|---|---|---|---|---|---|---|---|---|
| Obj. | 63107.01 | **74437.39** | 85178.56 | 122718.46 | Obj. | 32488.72 | **41375.67** | 48738.61 | 65628.95 |
| Time (s) | - | **0.58** | 2.19 | 5.60 | Time (s) | - | **0.04** | 0.23 | 0.18 |
| Gap (%) | - | **17.95** | 34.97 | 94.46 | Gap (%) | - | **27.36** | 50.02 | 102.01 |

## 4.4 ABLATION STUDY

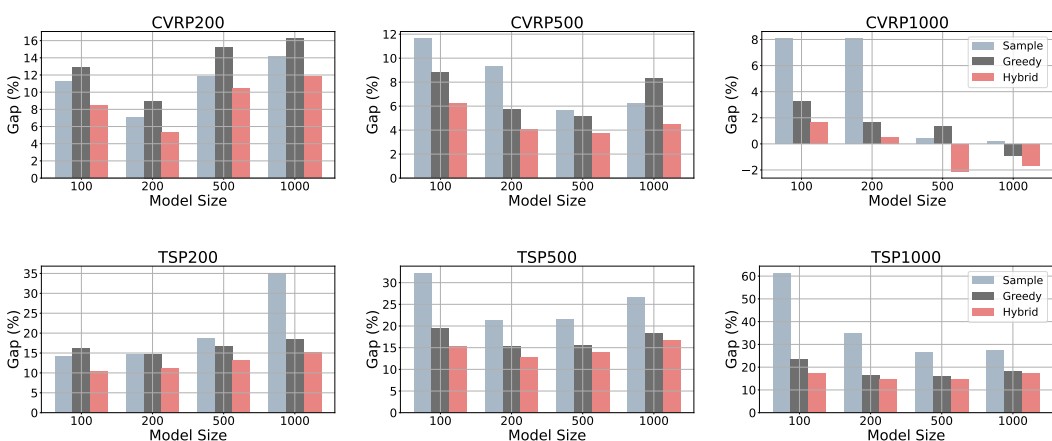

Figure 2: Comparison of the performance of greedy, sampling, and hybrid sampling strategies.

**Hybrid decoding**. We demonstrate the effectiveness of the hybrid decoding algorithm in our AGFN by comparing its performance against the pure sampling and greedy approaches. The results, presented in Figure 2, highlight the comparative advantages of the hybrid strategy across various scenarios. As shown, our hybrid method consistently achieves lower gap percentages than the other two methods across different problems and model sizes. These comparisons reveal that the hybrid method not only excels in generalization but also adapts more effectively to datasets of varying scales. Notably, in cases like CVRP1000 and TSP1000, the hybrid method significantly reduces the performance gap, demonstrating its robustness and versatility. These findings confirm that our hybrid models, trained on different node counts, outperform the individual sampling and greedy strategies, particularly when tested on large-scale instances.

**Local search in discriminator**. To train the discriminator in our AGFN, we employ local search to generate samples labeled as "true" for the discriminator. Here, we investigate the impacts of 1) using alternative methods for generating such true samples, and 2) removing the adversarial component. As illustrated for the scenario of CVRP200 in Figure 3, we exhibit the effects of using our local

search against the LKH (Lin-Kernighan-Helsgaun Helsgaun (2000)) heuristic to generate true samples during training. The results indicate that these two different search methods in the discriminator have almost equal impact on the performance. Regardless of the search method used, the generator converges in a similar yet satisfactory manner, indicating the robustness of the training process. Furthermore, we observe that incorporating the adversarial component into our AGFN significantly accelerates the convergence of the training curve and leads to superior overall performance. This is evident from the sharper decline in the average objective value when the adversarial component is included, compared to training without it. Note that, without the adversarial scheme, our model can be viewed as a simplified version of GFACS. The faster convergence and better results highlight the effectiveness of integrating adversarial training in our AGFN, making it a powerful tool for enhancing solution quality in complex optimization tasks such as CVRP.

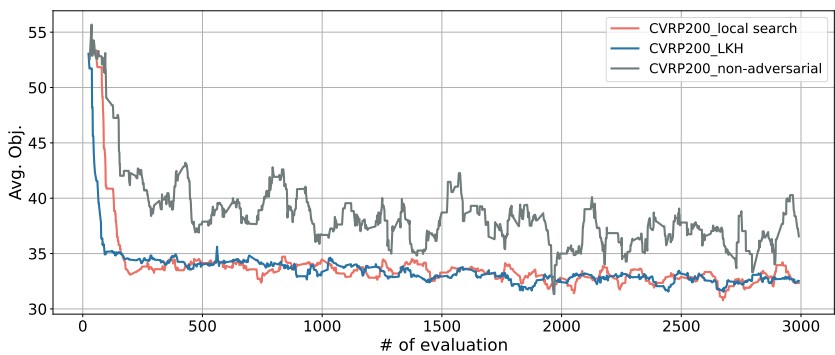

Figure 3: Training Performance Comparison of our model with different settings.

## 5 CONCLUSION

In this paper, we propose AGFN, a novel constructive framework for solving vehicle routing problems (VRPs). By leveraging GFlowNet and an adversarial training strategy, our approach provides high-quality solutions with strong generalization capabilities as a purely constructive neural solver. Extensive experimental comparisons with other representative Transformer-based constructive and improvement methods, as well as existing GFlowNet-based solvers, on both synthetic and real-world instances demonstrate the promise of AGFN in solving VRPs. We believe that AGFN can provide valuable insights and pave the way for further exploration of GFlowNets in solving more VRP variants and other combinatorial optimization problems. A limitation of AGFN is that its performance heavily relies on the generator's ability to produce diverse, high-quality solutions, which may account for its slightly lower test performance on 200-node CVRP and TSP instances. A potential future direction is to incorporate more advanced graph neural network (GNN) architectures with improved representation capability for VRPs, and more advanced adversarial training schemes. We will also compare AGFN against a broader range of robust neural VRP solvers on larger instances.

## 6 ACKNOWLEDGEMENT

This work is supported by the National Research Foundation, Singapore under its AI Singapore Programme (AISG Award No. AISG3-RP-2022-031), and the Singapore Ministry of Education (MOE) Academic Research Fund (AcRF) Tier 1 grant.

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

# A   PRELIMINARIES ON GFLOWNET

**GFlowNet** (Bengio et al., 2021) is a probabilistic framework designed to generate diverse structures or sequences by learning a distribution over a set of possible states $\mathcal{S}$, which are represented as a directed acyclic graph. GFlowNets have shown remarkable versatility across various challenging domains, including molecular discovery (Pan et al.), combinatorial optimization (Zhang et al., 2024), and recommendation systems (Liu et al., 2023). Let $s_0$ denote the initial state. A sequence of states, $\tau = (s_0 \rightarrow s_1 \rightarrow \cdots \rightarrow s_n)$, is generated via a policy that defines a probability distribution over actions at each state. The fundamental concept involves sampling sequences of actions that transition from an initial state $s_0$ to a terminal state $s_n$, with each sequence uniquely corresponding to an object $x \in \mathcal{X}$. In the context of VRP, $s_t$ represents the current partial route, covering a total of $t$ customers visited, while each $x$ corresponds to a unique, complete route that visits every customer exactly once and returns to the starting point.

The transition from a state $s_t$ to its child state $s_{t+1}$ is determined by an action $a_t$, sampled from a distribution $P_F(s_{t+1}|s_t)$, referred to as the *forward policy*. Conversely, the *backward policy*, denoted as $P_B(s_t|s_{t+1})$, captures the transition probability from a child state $s_{t+1}$ back to its parent state $s_t$. The marginal likelihood of sampling an object $x \in \mathcal{X}$ is defined as $P_F(x) = \sum_{\tau \in \mathcal{T}:\tau \rightarrow x} P_F(\tau)$, where $\tau \rightarrow x$ denotes a complete trajectory $\tau$ terminating at object $x$, and $P_F(\tau)$ represents the forward probability of a complete trajectory $\tau$. The principal aim of GFlowNet is to ensure that this marginal likelihood is proportional to the reward of the generated object while preserving the diversity of sequences: $P_F(x) \propto R(x)$.

**Trajectory Balance (TB)** (Malkin et al., 2022) is a widely adopted objective for training GFlowNets, designed to minimize the following loss function:

$$\mathcal{L}_{TB}(\tau; \boldsymbol{\theta}) = \left( \log \frac{Z_{\boldsymbol{\theta}} \prod_{t=0}^{n-1} P_F(s_{t+1}|s_t; \boldsymbol{\theta})}{R(x) \prod_{t=0}^{n-1} P_B(s_t|s_{t+1}; \boldsymbol{\theta})} \right)^2 . \tag{9}$$

The TB objective ensures that the product of forward transition probabilities aligns with the product of backward transition probabilities, thereby promoting consistent flow across all paths leading to the same outcome. The trajectory balance loss, $\mathcal{L}_{TB}$, comprises three key components: the source flow $Z_{\boldsymbol{\theta}}$, representing the initial state flow $F(s_0)$, which is computed as $Z_{\boldsymbol{\theta}} = \sum_{\tau \in \mathcal{T}} F(\tau)$; the forward policy $P_F(s_{t+1}|s_t; \boldsymbol{\theta})$; and the backward policy $P_B(s_t|s_{t+1}; \boldsymbol{\theta})$.

# B   HYPERPARAMETER IN HYBRID DECODING

As mentioned in Section 3.4, hyperparameter $\mathcal{P}$ is used to sample the node with the edge distribution probability $P_F(s_{t+1}|s_t; \boldsymbol{\theta}_{generator})$. Here, we evaluate the model performance under different value of $\mathcal{P}$. Based on the results shown in Table 4 and Table 5, We set the hyperparameter $\mathcal{P} = 0.05$ to achieve the highest solution quality.

Table 4: Sensitivity analyses of hyperparameters $\mathcal{P}$ in Hybrid Decoding on synthetic CVRP dataset.

| $\mathcal{P}$/Node | $|V| = 200$ | Gap(%) | $|V| = 500$ | Gap(%) | $|V| = 1000$ | Gap(%) |
|---|---|---|---|---|---|---|
| LKH-3(100) | 28.833135 | – | 66.902511 | – | 131.795858 | – |
| 0.01 | 31.555902 | 9.44 | 71.622154 | 7.05 | 134.074051 | 1.73 |
| 0.03 | 31.377661 | 8.83 | 71.111671 | 6.29 | **133.875488** | **1.58** |
| 0.05 | **31.260145** | **8.41** | **71.051109** | **6.20** | 133.966240 | 1.65 |
| 0.07 | 31.363468 | 8.78 | 71.092043 | 6.26 | 134.133682 | 1.77 |
| 0.10 | 31.347300 | 8.72 | 71.101418 | 6.28 | 134.436279 | 2.00 |

# C   GENERALIZATION ANALYSIS ON LARGER INSTANCES

To further evaluate the scalability of AGFN, we conducted experiments using the AGFN model trained on synthetic instances with 100 nodes and tested it on larger instances comprising 2,000,

Table 5: Sensitivity analyses of hyperparameters $\mathcal{P}$ in Hybrid Decoding on synthetic TSP dataset.

| $\mathcal{P}$/Node | $|V| = 200$ | Gap(%) | $|V| = 500$ | Gap(%) | $|V| = 1000$ | Gap(%) |
|---|---|---|---|---|---|---|
| LKH-3(100) | 10.719512 | – | 16.547770 | – | 23.123401 | – |
| 0.01 | 11.873071 | 10.76 | 19.172634 | 15.86 | 27.760757 | 20.05 |
| 0.03 | 11.854967 | 10.59 | 19.140463 | 15.67 | 27.882353 | 20.58 |
| 0.05 | **11.847540** | **10.52** | **19.076220** | **15.28** | **27.144802** | **17.39** |
| 0.07 | 11.867495 | 10.71 | 19.212893 | 16.11 | 27.231346 | 17.77 |
| 0.10 | 11.898133 | 11.00 | 19.295887 | 16.61 | 27.562878 | 19.20 |

Table 6: Comparative results on much larger synthetic CVRP dataset with up to 10,000 nodes. The 'Obj.' indicates the average total travel distance, while 'Time' denotes the average time to solve a single instance.

| CVRP | $|V| = 2000$ | | | $|V| = 3000$ | | | $|V| = 5000$ | | | $|V| = 10000$ | | |
|---|---|---|---|---|---|---|---|---|---|---|---|---|
| | Obj. | Gap (%) | Time (s) | Obj. | Gap (%) | Time (s) | Obj. | Gap (%) | Time (s) | Obj. | Gap (%) | Time (s) |
| LKH-3(1000) | 256.631797 | - | 224.98 | 383.820625 | - | 482.06 | - | - | - | - | - | - |
| AGFN-100 | **259.536438** | **1.13** | **2.61** | **369.998596** | **-3.60** | **4.03** | **607.290527** | **-** | **7.13** | **1146.165161** | **-** | **14.60** |
| GANCO-100 | 291.824432 | 13.71 | 6.39 | - | - | - | - | - | - | - | - | - |
| GFACS-200 | 284.539154 | 10.87 | 77.44 | 405.368347 | 5.61 | 139.31 | 663.644958 | - | 280.69 | - | - | - |
| POMO-100 | 627.439657 | 144.49 | 3.87 | 1124.936861 | 193.09 | 7.27 | 1507.168600 | - | 19.63 | - | - | - |
| POMO(*8)-100 | 411.848147 | 60.48 | 8.67 | 733.698417 | 91.16 | 20.33 | 1456.011373 | - | 45.38 | - | - | - |

3,000, 5,000, and 10,000 nodes. The 2,000 and 3,000 node scales contain 128 instances each, while the 5,000 and 10,000 node scales contain 64 instances each. For the baseline heuristic LKH-3, we limited its runtime to 30 minutes per instance.Due to time constraints, we were only able to provide LKH-3 results for 1,000 iterations and could not complete 10,000 iterations, as running 10,000 iterations for just 18 instances took more than 24 hours. The results, summarized in Table 6, demonstrate that our AGFN framework generalizes effectively to larger problem sizes, maintaining high solution quality and outperforming other baselines by a substantial margin across all CVRP instances. AGFN achieved the best objective values among neural models, showing gaps of 1.13% and -3.60% compared to LKH-3 on instances with 2,000 and 3,000 nodes, respectively. For larger instances with 5,000 and 10,000 nodes, AGFN completed each instance in just 14.60 seconds, whereas other baselines, including LKH-3, failed to produce results at this scale due to computational constraints. It is worth noting that our model, trained on 100-node instances, outperforms GFACS, which is pretrained on 200-node[1] instances, with much shorter computation time. Notably, on the largest instance with 10,000 nodes, AGFN completed in just 14.60 seconds for each instance, whereas other baselines including LKH-3 were unable to produce results for this scale due to computational constraints. These findings underscore the strong generalization ability and scalability of our approach, making it well-suited for solving real-world large-scale VRPs.

---

[1]We use the GFACS-200 because its pretrained model based on 100-node instances is unavailable.

