# OpenReview forum: "Adversarial Generative Flow Network for Solving Vehicle Routing Problems"
_ICLR.cc/2025/Conference — ICLR 2025 Poster_

### Official Review · Reviewer_ToYi · 2024-11-01

**Soundness:** 3
**Presentation:** 3
**Contribution:** 3
**Rating:** 6
**Confidence:** 2

**Summary:**

The paper titled "Adversarial Generative Flow Network for Solving Vehicle Routing Problems" introduces a novel framework, Adversarial Generative Flow Networks (AGFN), which leverages the capabilities of Generative Flow Networks (GFlowNets) and adversarial training to address the vehicle routing problem (VRP). The AGFN framework combines a generative model for constructing diverse solutions with a discriminative model for evaluating these solutions. The models are trained in an adversarial manner to enhance the overall solution quality. The authors apply AGFN to solve the capacitated vehicle routing problem (CVRP) and the traveling salesman problem (TSP), demonstrating its effectiveness against popular neural solvers and showcasing its strong generalization capabilities on both synthetic and real-world instances.

**Strengths:**

1） The paper presents a novel approach to VRPs by combining GFlowNets with adversarial training, which is a creative extension of existing methods.

2） The proposed AGFN framework addresses the scalability challenges faced by current neural solvers for VRPs.

3） The paper is well-written, with clear explanations of the AGFN framework and its components.

**Weaknesses:**

More analysis on the computational efficiency, especially in terms of scalability, would be beneficial.

**Questions:**

Hyperparameter Sensitivity: How sensitive is AGFN to hyperparameter tuning, particularly for the hybrid decoding method? Are there any guidelines or methods used to select these parameters?

---

### Official Review · Reviewer_bQyX · 2024-11-01

**Soundness:** 3
**Presentation:** 2
**Contribution:** 2
**Rating:** 6
**Confidence:** 3

**Summary:**

The article mainly uses the idea of ​​GFlowNet in CVRP problem that the solution can be inferred by probability. In order to prevent convergence to the local optimum, the author also introduced a discriminator to form an adversarial model. By alternately training the generator and the discriminator, GFlowNet can infer a high-level solution. In the experiment, the data sets of 200, 500, and 1000 were tested and good results were achieved.

**Strengths:**

GFlowNet was introduced in the CVRP problem to verify its feasibility and advantages in combinatorial optimization problems. In addition, adversarial training was added during the training phase, allowing the network to explore a wider area. Experiments have verified that this method can surpass some methods.
Avoiding local optima in the solution of combinatorial optimization problems is an important issue and the method given in the article is a good solution.
Very detailed experiments are carried out in the text and the results of these experiments are explained in detail, which will help to refer to future work.

**Weaknesses:**

The article mainly applies GFlowNet to the CVRP problem, but the corresponding improvements and innovations are slightly insufficient. In addition, the article does not go into details when describing the method, such as the explanation of the reward function and the meaning of the loss function.
The article mentions that generating high-quality diverse samples is the key to the effectiveness of the algorithm. Using adversarial learning to solve this problem is a common idea. We do not see that the author has made any significant improvements to this framework to improve its ability to generate diverse samples.
The code is currently not open-source and cannot be evaluated for the reproducibility of the experiments.

**Questions:**

From the experiment, the method in this paper is better than the traditional reinforcement learning autoregressive method in solving path planning problems, but how is the effect on a larger scale?(like 5000, 10000) Whether the algorithm will fail on larger datasets？
The design of ​​gflownet makes this method well adapted to problems with multiple optimal solutions, but for problems such as CVRP, the optimal solution is unique, so what is the advantage of using gflownet?
Because adversarial learning is the core of this paper, we hope that the author can show from a theoretical or experimental perspective that it has advantages over other methods of generating diverse samples. Or the author can prove the novelty of this paper by showing that they have made improvements to the adversarial learning framework.

---

### Official Review · Reviewer_opH2 · 2024-11-04

**Soundness:** 3
**Presentation:** 3
**Contribution:** 2
**Rating:** 6
**Confidence:** 2

**Summary:**

This paper introduces a novel neural solver for vehicle routing problems, specifically CVRP and TSP. It employs a generative flow network (GFlowNet) to learn a diverse set of potential solution trajectories to address the common challenge in VRP solvers of falling into locally optimal solution. The GFlowNet is trained in an adversarial manner where a discriminator is used to distinguish between the raw outputs of the network and their corresponding optimized trajectories obtained via local search techniques. To further balance the exploration and exploitation, the method uses a hybrid decoding strategy, similar to $\epsilon$-greedy. Experimental results show that the proposed method generally outperforms existing baselines, especially on larger problem instances with higher numbers of nodes.

**Strengths:**

- Leveraging GFLowNet and the proposed hybrid decoding strategy is well-suited for generating a diverse set of trajectories, which could help to avoid local optima trajectories.
- The proposed method shows good performance and scalability. It works especially well in larger instances with more nodes compared with the baselines.

**Weaknesses:**

- The paper primarily integrates existing components such as the GFlowNet, adversarial training, and $\epsilon$-greedy, into a coherent system for solving vehicle routing problems.
- A more detailed explanation and discussion could be beneficial. For example, how to generate the sparse edge set? How to pick the optimal value of $\mathcal{P}$ for the hybrid decoding?

**Questions:**

Please see Weaknesses

---

### Meta-Review · Area_Chair_TdpD · 2024-12-17

**Metareview:**

This paper proposes the Adversarial Generative Flow Network (AGFN) for vehicle routing problems. It combines GFlowNet with adversarial training and hybrid decoding to generate diverse solutions and avoid local optima. Experimental results show it outperforms baselines in solving CVRP and TSP, with good generalization and scalability.

The strengths include its innovative combination of methods, strong performance on larger instances, and detailed experiments. Weaknesses are a lack of in-depth explanation in some areas and questions about its performance on very large datasets. However, the authors' active response during the rebuttal, such as providing more details and conducting additional experiments, addressed many concerns. Overall, the paper's novelty and potential contribution to the field of vehicle routing problems justify its acceptance, as the improvements made during the review process enhance the clarity and credibility of the proposed approach.

**Additional Comments On Reviewer Discussion:**

During the rebuttal period, reviewers raised several points. These included asking for more details on aspects like generating the sparse edge set, selecting optimal hyperparameters for hybrid decoding, showing the effect on larger scale datasets, clarifying the advantage of using GFlowNet, improving explanations of the reward and loss functions, and demonstrating the novelty of the adversarial learning framework. The authors addressed these by adding detailed descriptions in the manuscript, conducting additional experiments and presenting the results, providing theoretical and practical justifications, and making the code and related data publicly available. In weighing these points, the authors' responsiveness and the demonstrated improvements to address the concerns, along with the core novelty and good performance of the AGFN framework, led to the decision to accept the paper.

---

### Decision · Program_Chairs · 2025-01-22

Accept (Poster)